# Carbon-Ion Radiotherapy Combined with Concurrent Chemotherapy for Locally Advanced Pancreatic Cancer: A Retrospective Case Series Analysis

**DOI:** 10.3390/cancers15102857

**Published:** 2023-05-22

**Authors:** Masahiko Okamoto, Shintaro Shiba, Daijiro Kobayashi, Yuhei Miyasaka, Shohei Okazaki, Kei Shibuya, Tatsuya Ohno

**Affiliations:** 1Heavy-Ion Medical Center, Gunma University, 3-39-22, Showa-machi, Maebashi 371-8511, Gunma, Japan; shiba4885@yahoo.co.jp (S.S.); dkobayashi@gunma-u.ac.jp (D.K.); y.miyasaka@gunma-u.ac.jp (Y.M.); o_syohei_1015@yahoo.co.jp (S.O.); shibukei@gunma-u.ac.jp (K.S.); tohno@gunma-u.ac.jp (T.O.); 2Department of Radiation Oncology, Gunma University Graduate School of Medicine, 3-39-22, Showa-machi, Maebashi 371-8511, Gunma, Japan; 3Department of Radiation Oncology, Shonan Kamakura General Hospital, 1370-1, Okamoto, Kamakura 247-8533, Kanagawa, Japan; 4Department of Radiation Oncology, Gunma Prefectural Cancer Center, 617-1, Takabayashi-nishi, Ota 373-8550, Gunma, Japan

**Keywords:** carbon ion radiotherapy, pancreatic cancer, particle beam radiotherapy, heavy-ion radiotherapy, locally advanced pancreatic cancer

## Abstract

**Simple Summary:**

The effectiveness of carbon-ion radiotherapy (CIRT) with systemic chemotherapy for locally advanced pancreatic cancer was examined retrospectively. Of the 44 patients receiving CIRT, 37 patients also received neoadjuvant chemotherapy, and all patients received concurrent and adjuvant chemotherapy. The median survival time of all patients was 34.5 months after the initial treatment and 29.6 months after the initial day of CIRT. The estimated two-year overall survival rate and local control after CIRT was 56.6% and 76.1%, respectively. Even in the era of multiagent chemotherapy, local treatment with CIRT has a survival benefit for local advanced unresectable pancreatic cancer.

**Abstract:**

Systemic chemotherapy has significantly improved in recent years. In this study. the clinical impact of carbon-ion radiotherapy (CIRT) with concurrent chemotherapy for locally advanced unresectable pancreatic cancer (URPC) was evaluated. Methods: Patients with URPC who were treated with CIRT between January 2016 and December 2020 were prospectively registered and analyzed. The major criteria for registration were (1) diagnosed as URPC on imaging; (2) pathologically diagnosed adenocarcinoma; (3) no distant metastasis; (4) Eastern Cooperative Oncology Group performance status of 0–2; (5) tumors without gastrointestinal tract invasion; and (6) available for concurrent chemotherapy. Patients who received neoadjuvant chemotherapy (NAC) for more than one year prior to CIRT were excluded. Results: Forty-four patients met the inclusion criteria, and thirty-seven received NAC before CIRT. The median follow-up period of living patients was 26.0 (6.0–68.6) months after CIRT. The estimated two-year overall survival, local control, and progression-free survival rates after CIRT were 56.6%, 76.1%, and 29.0%, respectively. The median survival time of all patients was 29.6 months after CIRT and 34.5 months after the initial NAC. Conclusion: CIRT showed survival benefits for URPC even in the multiagent chemotherapy era.

## 1. Introduction

The standard treatment for locally advanced unresectable pancreatic cancer (URPC) is chemotherapy or chemoradiation therapy (CRT). However, since the pancreas is surrounded by radiosensitive organs, such as the stomach, duodenum, liver, kidney, and intestine, high-dose X-ray irradiation is sometimes difficult to achieve even with high-precision treatment technologies. Hence, long-term local tumor control after conventional X-ray radiation therapy (XRT) for locally advanced URPC is insufficient. In a meta-analysis of 15 randomized controlled trials (1128 patients) comparing CRT with radiation therapy or chemotherapy alone, CRT was associated with superior survival at 6 and 12 months compared with radiation therapy or chemotherapy alone, but the difference was not significant at 18 months [1]. Even with high-precision radiotherapy, such as intensity-modulated radiotherapy, the median overall survival (OS) was 14–17 months, and the two-year local control (LC) rate was 39–42% [2,3,4].

On the other hand, in the last two decades, carbon-ion radiation therapy (CIRT) has emerged as a potential and promising treatment for cancers refractory to conventional X-ray radiotherapy. Carbon-ion (C-ion) beams have two features. First, because CIRT beams have a Bragg peak, they can be stopped at the distal end of tumors, reducing the irradiated volume of healthy organs and increasing the treatment effect ratio. Second, CIRT beams with high linear energy transfer have a relative biological effectiveness (RBE) about three times higher than that of X-rays when compared at the same physical dose [5,6,7].

Several studies reported the efficacy and safety of CIRT for locally advanced pancreatic cancer [8,9]. In 2018, a multicenter, retrospective study conducted in Japan analyzed 72 patients who underwent CIRT for locally advanced pancreatic cancer between 2012 and 2014, showing a two-year OS rate of 46%, a median survival time (MST) of 21.5 months, and a cumulative local recurrence rate of 24% two years after CIRT [9]. Another single-center phase 1/2 prospective clinical trial of CIRT with gemcitabine (GEM) for locally advanced pancreatic cancer showed a two-year survival rate of 48% in patients who received more than 45.6 Gy [8]. However, these survival rates after CIRT, although superior to those after conventional XRT, are still unsatisfactory.

Meanwhile, in recent years, multiagent combination chemotherapy, such as nab-paclitaxel combined with GEM and FORFILINOX (combination of oxaliplatin, irinotecan, fluorouracil, and leucovorin), showed improved clinical results in patients with metastatic [10,11,12] and locally advanced pancreatic cancer [13,14,15,16]. Here, we analyzed the clinical outcomes of CIRT for locally advanced pancreatic cancer in the multiagent chemotherapy era.

## 2. Materials and Methods

### 2.1. Patient Eligibility

In Japan, all patients with locally advanced URPC have been prospectively enrolled in the national registry system since 2016. We reviewed the medical records of patients with locally advanced pancreatic cancer treated with CIRT at the Gunma University Heavy Ion Medical Center (GHMC) between January 2016 and December 2020. The eligibility criteria of CIRT were as follows: (1) diagnosed as locally advanced URPC by radiological imaging; (2) pathologically diagnosed adenocarcinoma; (3) no distant metastasis, except for para-aortic lymph nodes that could be included in the irradiation target volume; (4) Eastern Cooperative Oncology Group performance status of 0–2; (5) tumors without gastrointestinal tract invasion; (6) adequate organ function for concurrent chemotherapy; (7) ages of 20–80 years old; and (8) informed consent for the treatment. Patients were excluded if they: (1) had an active stomach and/or duodenal ulcer; (2) had an active infection in the upper abdomen; (3) underwent neoadjuvant chemotherapy (NAC) exceeding one-year duration; (4) had a metallic stent inserted to treat obstructive jaundice; or (5) were enrolled in another multicenter prospective study. Metallic stents have a considerable impact on the delivery distance of C-ions; therefore, in metal stent implantation cases, CIRT was performed after replacement with a plastic stent. The treatment protocol of the current study was reviewed and approved by the Gunma University Ethics Committee of Human Clinical Research (approved code: 1108 and 1370), and all patients signed an informed consent form before the initiation of therapy.

### 2.2. Carbon-Ion Radiotherapy

C-ion beams with energy of 290 MeV/u, 380 MeV/u, and 400 MeV/u were generated by a heavy-ion accelerator at GHMC and selected according to the tumor depth. An XiO-N system (version 4.47; collaborative product of Elekta AB, Stockholm, Sweden, and Mitsubishi Electric, Tokyo, Japan) was used for treatment planning. This system incorporates a dosing engine for ion-beam RT (K2dose). For clinical radiation dose in CIRT, we used the RBE-weighted dose in Gy units, which was calculated by multiplying the physical dose with the RBE of C-ion beams. The mixed-beam model based on the survival of tumor cells in the human salivary gland was used as the RBE model in this study [17]. Before C-ion RT, patients were immobilized using tailor-made fixation cushions and thermoplastic shells to acquire treatment planning CT images; then, respiratory-gated and four-dimensional CT images and dynamic contrasted CT images were acquired. Patients received C-ion RT once daily, four days a week (Tuesday–Friday), under respiratory gating. One port of fixed beamline was treated in each session. All patients fasted for ≥3 h before undergoing CIRT.

### 2.3. Treatment Planning

Treatment planning CT images were merged with dynamic contrast-enhanced CT and magnetic resonance imaging and FDG-PET CT images to delineate the GTV. CTV was defined as GTV with a margin of at least 5 mm in all directions and at least 1 cm toward the long axis of the pancreas. Then, CTV was expanded to include the prophylactic lymph node area and the neuroplexus region. We selected the celiac, superior mesenteric, peripancreatic, portal, and para-aortic regions cephalad to the transverse portion of the duodenum as prophylactic lymph node areas. The planning organ at risk volume (PRV) was created by adding a 2 mm margin to the gastrointestinal tract. CTV regions that overlapped with the PRV of the gastrointestinal tract were removed from CTV. Planned target volume (PTV) was defined as CTV with a margin of 3 mm to cover possible positioning errors. When PTV overlapped with an organ at risk, the area was reduced accordingly. The maximal irradiated doses of the stomach and duodenum were restricted to <45 Gy. Stomach and duodenum volumes irradiated with >30 Gy were limited to <10 cm^3^. The maximum irradiation dose to the spinal cord was restricted to <30 Gy. For the liver, the volume irradiated with a dose of >20 Gy was limited to <35% of the total volume. For the kidney, the volume irradiated with a dose of >20 Gy was limited to <50% of the volume of each kidney. We did not set dose constraints for the bile ducts as they are difficult to avoid in cases of pancreatic head cancer. However, we strongly recommend avoiding the placement of large-diameter metallic stents to prevent serious adverse events after CIRT. Dose constraints are summarized in Table 1. The portal vein in the pancreatic portion is often obstructed by the tumor itself, but this is compensated by the development of collateral vessels. Thus, if stenosis occurs in this area after CIRT, it rarely causes serious liver damage. Therefore, we did not set dose constraints for the portal vein or use portal vein stenosis as an ineligibility criterion. In a standard case, the prescribed dose was 55.2 Gy in 12 fractions. The dose distribution of CIRT for a typical case is shown in Figure 1. We always perform a second respiratory-gated CT scan a day before the start of CIRT to confirm the dose distribution to the GTV and risk organs, and, if necessary, to make a new treatment plan. At the time of irradiation, the position of the target and risk organs were matched using fluoroscopy or in-room CT, and if there was a large displacement, a new treatment plan was created based on the new CT.

### 2.4. Chemotherapy

All patients received concurrent chemotherapy with either GEM or Tegafur-Gimeraci-Oteracil potassium (TGO). As for the selection policy of concurrent chemotherapy, patients treated with a GEM-based regimen for NAC received GEM, while patients treated with either FORFIRINOX or TGO for NAC who were unable to continue GEM received TGO. GEM was administrated at a standard dose of 1000 mg/m^2^ on days 1, 8, and 15 of the CIRT treatment. TGO was administered orally twice daily from day 1 to day 28 of CIRT. The TGO dose was defined by body surface area (BSA): 80 mg/day for BSA < 1.25 m^2^, 100 mg/day for BSA of 1.25–1.5 m^2^, and 120 mg/day for BSA ≥ 1.5 m^2^. If the dose or schedule of GEM or TGO was reduced at NAC, the dose was adjusted to match the reduction. The criteria for chemotherapy administration were set as follows: white blood cell count ≥ 2000 cells/mm^3^, neutrophil count ≥ 1000 cells/mm^3^, platelet count ≥ 75,000/mm^3^, blood bilirubin level < 3.0 mg/dL, serum aspartate aminotransferase and alanine aminotransferase levels < 200 IU/L, and serum creatinine level < 1.5 mg/dL. There were no specific protocols for adjuvant chemotherapy after CIRT. However, we recommended at least six months of adjuvant chemotherapy or more if there were no problems with adverse events.

### 2.5. Evaluation and Follow-Up

The follow-up policy was to evaluate patients at least every three months for a minimum of two years. During the follow-up, CT and/or FDG-PET scans, tumor markers (CEA and CA19-9), physical examination, and interviews were performed. Acute and late toxicities were graded according to the National Cancer Institute Common Toxicity Criteria (version 4.0). The highest grade within three months and after three months was evaluated as acute and late toxicity, respectively. Tumor response was evaluated by comparing pre- and post-treatment CT scans and FDG-PET scans. Even after the patients developed metastases, we continued to collect information on local tumor control as long as they were alive.

### 2.6. Statistical Analysis

LC, PFS, and OS rates were calculated using the Kaplan–Meier method. Local recurrence was defined as recurrence in the pancreatic region, including the lymph nodes or outside of the irradiated field, based on CT or FDG-PET scans. LC was defined by the absence of local recurrence, as indicated by no evidence of increase in tumor size by >20% on CT and no accumulation on FDG-PET at >6 months after treatment. If FDG accumulation increased in the tumor with or without increase in size, it was considered to be progressive. All endpoints were calculated starting from the first day of CIRT, except for the analysis annotated as the period from the first day of NAC. Univariate analysis was performed, and log-rank test was used to compare the survival distributions in the two groups. To calculate the cutoff values for continuous variables, the receiver operating characteristic curve analysis was used. Pearson’s chi-square test was used to compare the observed frequencies between the two groups. Student’s *t*-test was performed to compare the means between the two groups. A *p*-value of <0.05 was considered statistically significant. All statistical analyses were conducted using IBM SPSS Statistics for Mac, version 26.0 (SPSS, Armonk, NY, USA).

## 3. Results

### 3.1. Patient Characteristics

From January 2016 to December 2020, 44 patients met the inclusion criteria and were treated with CIRT. The patient characteristics are summarized in Table 2. All patients were diagnosed with T4 URPC. Thirty-seven patients received NAC before CIRT. The median duration of NAC was 88 days, and in 16 patients, its duration was longer than 100 days. All patients received a CIRT of 55.2 Gy in 12 fractions. All patients also received concurrent chemotherapy with CIRT, including GEM in 26 patients and TGO in 18 patients. All patients received adjuvant chemotherapy after CIRT, and 27 patients used a multidrug regimen. There was no specific duration limit for adjuvant chemotherapy.

### 3.2. Survival and LC Rate

The median follow-up period of living patients was 26.0 (6.0–68.6) months after CIRT. The MST of all patients after CIRT was 29.6 months. The 2-year OS, LC, and progression-free survival (PFS) rates were 56.6%, 76.1%, and 29.0%, respectively (Figure 2a). Since the start of the initial treatment, i.e., the start of NAC or CIRT in cases not receiving NAC, the MST was 34.5 months (Figure 2b). As for the first relapse pattern after CIRT, 26 out of 32 cases had distant metastasis, and 6 cases had local relapses. During the whole course of the study, in- and out-field recurrences were confirmed in 6 and 2 of the 44 patients, respectively. Among eight local recurrence cases, six cases with no evidence of distant metastasis at the time of recurrence were treated with repeated CIRT as a salvage treatment. The results of univariate analysis with the endpoint of OS are summarized in Table 3. Age and serum CA19-9 levels were the significant risk factors for OS.

The results of the univariate analysis of risk factors for in-field local recurrence are summarized in Table 4. The GTV D95 dose was the only significant risk factor. Moreover, in patients who received adjuvant chemotherapy with multiple agents, LC tended to be better (*p* = 0.079).

### 3.3. Toxicities

Adverse events that occurred after CIRT are summarized in Table 5. Acute grade 2 ulcers in the upper gastrointestinal tract were found in four patients within three months after CIRT. One acute upper gastrointestinal ulcer continued after three months and was also counted as a late adverse event, and two other cases of late grade 2 or higher upper gastrointestinal ulcers were observed. One grade 3 ulcer was a case of duodenal ulcer perforation at seven months after CIRT, although local tumor regrowth was also observed, which may have been caused by the recurrent tumor. The gastric volume irradiated with a dose of >30 Gy was 5.2 ± 0.5 cc in the grade 2 or higher ulcer group, whereas it was 4.1 ± 1.4 cc in the grade 0–1 group, with no significant difference between the two groups (*p* = 0.435). Acute and late grade 2 or higher lower gastrointestinal disorders were observed in two and three patients, respectively, and all patients had diarrhea. Five cases of grade 2 or higher biliary stricture were identified during the late phase after CIRT, and three of them were grade 3 ulcers. Moreover, there were no treatment-related deaths.

Regarding acute myelosuppression, 20 of 26 patients in the GEM group had grade 2 or higher leukopenia, compared to 4 of 18 patients in the TGO group, which was significant (*p* < 0.001). In addition, 8 of 26 patients in the GEM group had grade 2 or higher thrombocytopenia, whereas no patients in the TGO group had grade 2 or higher thrombocytopenia (*p* < 0.01).

## 4. Discussion

Recently, favorable clinical outcomes have been reported using new chemotherapy combination regimens. For locally advanced pancreatic cancer, gemcitabine plus nab-pacritaxel or FOLFIRINOX has been shown to achieve a favorable MST of 14–22.5 months [13,14,15,16]. On the other hand, chemoradiotherapy using X-ray does not show an additional benefit in survival over chemotherapy alone [2,3,4]. One reason for this could be that X-ray irradiation, even with high-precision radiotherapy, such as IMRT, is not sufficient to eliminate adenocarcinomas, which often show radioresistance; moreover, most patients diagnosed with URPC may already have microscopic distant metastases at the time of radiotherapy. C-ion beams have high LET and could overcome this radioresistance. A retrospective study of URPC cases treated with C-ion from 2012 to 2014 reported an MST of 21.5 months [9]. Although this is longer than conventional CRT, it is almost similar to that of recent chemotherapy-alone studies. In our data, the MST was 29.6 months after CIRT and 34.5 months after NAC. It is difficult to directly compare these data because our data do not include patients with aggressive progression who develop distant metastases early after the start of chemotherapy. Nonetheless, our data suggest the possibility of prolonged survival with the addition of CIRT over chemotherapy alone when combined with latest chemotherapy regimens. Our data also showed no significant survival difference between NAC durations above or below 100 days. This suggests that there is no limited timing for the addition of CIRT, and its addition at any time during systemic chemotherapy is effective. As for the concurrent chemotherapy regimen, there was no significant difference in survival between GEM and TGO. However, due to the limited number of patients, further study is necessary to compare their efficacy and safety.

The results of the univariate analysis of OS showed that younger age (<70 years) and lower serum CA19-9 (<150 ng/mL) levels at the time of treatment were significant favorable prognostic factors. In addition, in patients who received adjuvant chemotherapy with multiple agents, a prolonged OS was observed, although not significant. As this was a prospective registry study and most patients received adjuvant chemotherapy after CIRT at other referral hospitals, the information on the precise treatment duration of adjuvant chemotherapy in our cohort could not be obtained. However, younger patients are generally considered more likely to receive longer adjuvant chemotherapy. Controlling potential systemic diseases with multidrug chemotherapy while adding a powerful CIRT to the primary site with very large numbers of tumor cells may be an ideal treatment strategy. On the other hand, CA19-9 is also a significant prognostic factor for systemic chemotherapy [18,19] and surgery [20,21]. Generally, tumor markers reflect tumor aggressiveness, suggesting that patients with high CA19-9 may be at a higher risk for potential distant metastasis. It is suggested that appropriate case selection and post-treatment chemotherapy are important in combination with CIRT.

In the present study, local recurrence was observed in eight cases during the entire course, of which six cases were recurrences within the irradiated field. A GTV-D95 of <52.8 Gy was identified as a risk factor for in-field recurrence. Therefore, in cases where GTV was inadequately covered in relation to OAR, it was considered necessary to perform procedures, such as spacer insertion, to keep the OAR away from GTV. Moreover, in three cases, local recurrence occurred after a long period of time, such as 20 months after CIRT, and all of these recurrences were within the irradiated fields. This suggests that a dose of 55.2 Gy may not be sufficient to completely eliminate cancer cells in some cases. Recently, the safety of CIRT re-irradiation for local recurrence after CIRT in URPC has been reported [22,23]. Thus, we treated local recurrences without distant metastases with salvageable CIRT re-irradiation whenever possible. In fact, CIRT re-irradiation was performed in six cases in the current study, which may have contributed to the improved long-term survival. However, re-irradiation has risks of adverse events and cost issues. Therefore, dose escalation studies aiming to improve the LC rate with a single course of treatment are necessary.

In our study, late upper gastrointestinal disorders of grade 3 or higher were identified in only 1 of the 44 patients. Grade 3 biliary tract disorders were identified in three cases, which was similar in frequency to previous CIRT reports [8,24], even after six cases of re-irradiation. This could indicate that our treatment strategy works well. However, in CIRT for the pancreas, it is necessary to manage the intra- and interfraction movements of the GTV and risk organs. Recently, plan optimization with scheduled multiple CT imaging for re-evaluation has been reported to be useful in overcoming the intra- and interfraction movements [25]. We currently perform single re-evaluation CT a day before treatment initiation, but we are considering the use of multiple CT routinely as described in this report. On the other hand, the range of C-ion beams varies greatly with changes in densities before the target. For example, if the amount of air between the skin and the target increases, the beam penetrates the target and then stops, resulting in an increase in OAR dose on the distal side of the target. This uncertainty in beam range could affect not only the OAR but also the target coverage, which could lead to worse treatment outcomes. Therefore, during treatment, it is necessary to pay attention to the accuracy of the target location and to changes in surrounding gases. If the amount of gas differs significantly from the initial treatment plan, a second treatment plan should be created.

Our study had some limitations. First, this was not a prospective interventional study, so chemotherapy before and after CIRT was not uniform. Second, this was a single-center study and did not include a large number of patients. Third, the results may be affected by a selection bias that excludes aggressive cases that develop distant metastases early after the start of chemotherapy due to the inclusion of NAC. However, the results of this study are promising, and we plan to extend the follow-up period with more cases for further analysis.

## 5. Conclusions

Although the chemotherapy before and after CIRT was not uniform, the two-year survival rate of 56.6% and MST of 29.6 months after CIRT are considered promising after a median follow-up period of 26.0 months. Controlling potential systemic diseases with multidrug chemotherapy before and after CIRT while adding a strong CIRT to primary tumors with very large numbers of tumor cells could be an ideal treatment strategy. We will continue to increase the number of cases and extend the follow-up period to analyze more optimal chemotherapy regimens and durations combined with CIRT.

## Figures and Tables

**Figure 1 cancers-15-02857-f001:**
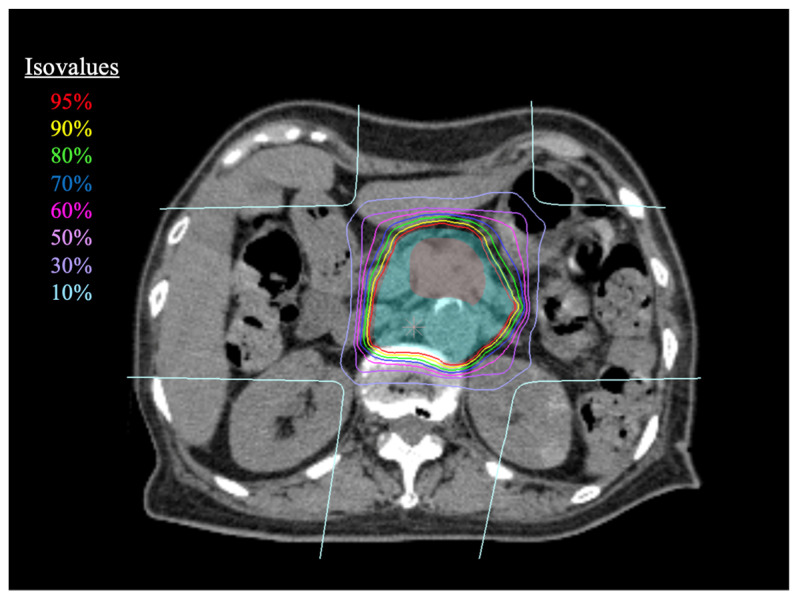
Dose distribution of a typical case. The prescribed dose was 55.2 Gy in 12 fractions. Color wash structures indicate gross tumor volume (red) and clinical target volume (cyan).

**Figure 2 cancers-15-02857-f002:**
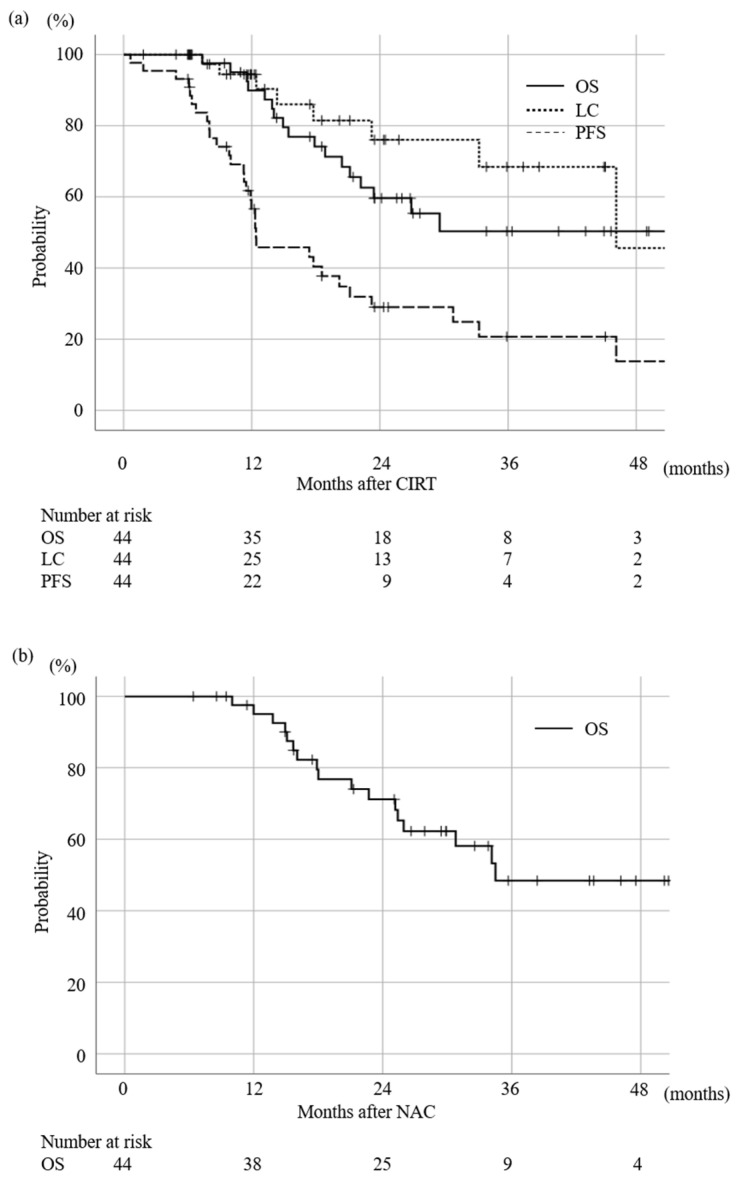
Survival rate after the initial day of (**a**) CIRT and (**b**) start of initial treatment (NAC (*n* = 37), CIRT (*n* = 7)). The survival rate was calculated by Kaplan–Meier methods. OS: overall survival rate, LC: local control rate, PFS: progression free survival.

**Table 1 cancers-15-02857-t001:** Dose constraints for organs at risk.

Structure	Constraints
Stomach, Duodenum	Max dose < 45 Gy; volume receive more than 30 Gy < 10 cm^3^
Spinal cord	Max dose < 30 Gy
Liver	Volume received more than 20 Gy < 35%
Kidney	Volume received more than 20 Gy < 50% of each kidney

**Table 2 cancers-15-02857-t002:** Patient characteristics.

Patient Characteristics		Value
Age (years)	median (range)	68 (44–79)
Sex	male/female	30:14
Tumor location	head/body	17:27
Tumor to GI tract distance (mm)	median (range)	3 (0–24)
N stage	N0:N1	36:8
Neoadjuvant CT	multi agent/single/none	35:2:7
NAC regimen	GnP/FFX/GEM/GEM + TGO/TGO	29:5:1:1:1
NAC duration (days)	median (range)	88 (31–343)
Concurrent CT	GEM:TGO	26:18
Adjuvant CT	multi agent/single	27:17
AC regimen	GnP/FFX/GEM/GEM + TGO/TGO	23:3:7:1:10

CT: chemotherapy, NAC: neoadjuvant chemotherapy, GnP: gemcitabine with nab-pacritaxel, FFX: FORFIRINOX, GEM: gemcitabine, TGO: Tegafur-Gimeracil-Oteracil potassium, AC: adjuvant chemotherapy.

**Table 3 cancers-15-02857-t003:** Univariate analysis for overall survival rate.

Factor	Category	Number	95% CI of mOS (Month)	*p*-Value
Age	≥70 <70	16 28	17.5–32.2 40.1–60.5	0.038
Sex	Male Female	30 14	30.4–43.3 20.0–47.9	0.069
Tumor location	Head Body	17 27	19.8–33.4 36.6–58.1	0.169
CA19-9 at CIRT	≥150 (ng/mL) <150 (ng/mL)	11 33	12.7–24.0 39.6–59.1	0.001
Concurrent CT	GEM TGO	26 18	29.3–43.4 25.8–51.1	0.436
Neoadjuvant CT	Single agent Multiple agents None	2 35 7	23.4–23.4 29.7–41.0 6.0–55.9	0.406
NAC duration	≥100 days <100 days None	16 21 7	24.3–42.5 28.4–42.8 6.0–55.9	0.449
Adjuvant CT	Single agent Multiple agents	17 27	21.6–46.4 32.1–45.0	0.077
Tumor to GI distance	≥3 mm <3 mm	28 16	30.1–51.8 26.6–41.9	0.555

CT: chemotherapy, CIRT: carbon-ion radiotherapy, NAC: neoadjuvant chemotherapy, GEM: gemcitabine, TGO: Tegafur-Gimeracil-Oteracil potassium, GI: gastrointestinal, mOS: median overall survival duration.

**Table 4 cancers-15-02857-t004:** Univariate analysis for in-field local control rate.

Factor	Category	Number	95% CI of mLC (Month)	*p*-Value
Age	≥70 y.o <70 y.o	16 28	38.4–56.2 33.1–64.9	0.390
Tumor location	Head Body	17 27	NIA * 35.4–63.5	0.122
CA19-9 at CIRT	≥150 (U/mL) <150 (U/mL)	11 33	28.3–49.9 40.3–67.4	0.551
Concurrent CT	GEM TGO	26 18	36.5–52.2 33.7–69.5	0.857
Tumor to GI distance	≥3 mm <3 mm	28 16	34.5–66.8 31.8–54.1	0.878
Tumor diameter	≥30 mm <30 mm	33 11	48.8–70.7 26.3–54.7	0.603
GTV D95	≥52.8 Gy <52.8 Gy	30 14	56.3–74.1 16.3–46.1	0.015
Adjuvant CT	Single agent Multiple agents	17 27	31.9–66.8 36.5–45.9	0.079

*: could not calculate because there is no in-field recurrence in this group. CT: chemotherapy, CIRT: carbon-ion radiotherapy, GTV: gross tumor volume, mLC: median local control duration, GEM: gemcitabine, TGO: Tegafur-Gimeracil-Oteracil potassium.

**Table 5 cancers-15-02857-t005:** Numbers of cases for each acute and non-hematological late adverse events after CIRT.

Acute Toxicity					
	**Adverse event**	**Grade 0–1**	**Grade 2**	**Grade 3**	**Grade 4–5**
	Upper GI tract	40	4	0	0
	Lower GI tract	42	2	0	0
	Biliary tract	41	3	0	0
	Dermatitis	44	0	0	0
	Nausea	40	4	0	0
**GEM**	Leukopenia	6	11	9	0
	Thrombocytopenia	18	2	6	0
**TGO**	Leukopenia	14	3	1	0
	Thrombocytopenia	18	0	0	0
**Late toxicity**					
**Adverse event**		**Grade 0–1**	**Grade 2**	**Grade 3**	**Grade 4–5**
**Upper GI tract**		41	2	1	0
**Lower GI tract**		41	3	0	0
**Biliary tract**		39	2	3	0
**Dermatitis**		44	0	0	0

GI: gastrointestinal, GEM: gemcitabine, TGO: Tegafur-Gimeracil-Oteracil potassium.

## Data Availability

The data presented in this study are available on request from the corresponding author. The data are not publicly available due to privacy and ethical restrictions.

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
