# Peer review of "Carbon-Ion Radiotherapy Combined with Concurrent Chemotherapy for Locally Advanced Pancreatic Cancer: A Retrospective Case Series Analysis"

_cancers, 2023, doi:10.3390/cancers15102857_

Round 1
Reviewer 1 Report
Thank you for the opportunity to review this original research.
I think that the topic can be of interest to the journal’s readers. However, the manuscript must be improved.
In particular:
-Patient eligibility:
1) please specify what is the minimum distance between the tumor and gastrointestinal tract
2) please specify if portal thrombosis is an exclusion criterion. If not, please detail what is the risk of liver failure due to portal thrombosis considering the recent data both in pancreatic cancers and HCC who underwent HCC. (doi: 10.1002/cncr.27723; Kasuya, doi: 10.18632/oncotarget.27028)
3) please specify why metal stent is an exclusion criterion
- Carbon ion radiotherapy
1) please detail the radiobiological model used
2)do you use fixed beamlines or a gantry?
3) please specify the optimization of treatment that you used ( robust optimisation?
4) How was the planned delivery gated?
5) How did you check and also manage the organ motion and the inter/intra-fraction anatomical variation? (doi.org/10.1016/j.radonc.2022.09.005)
- Treatment planning:
1) please define the lymph node areas according to the localization of the tumour and the guidelines used
2) please provide a table of the constraints used during the planning. In particular, considering the liver toxicity reported in CIRT literature I do think it is of utmost importance to specify the constraints used for portal vein, vessels, liver, and biliary tracts as well as for gastro-intestinal structures (colon, small bowel, duodenum, stomach…)
-Chemotherapy
1) Please specify the criteria for which a patient received FORFIRINOX or GEM
-Toxicities:
1) You reported “ acute grade 2 or higher ulcers were found in 4 patients”, but the table reported 4 G2 and no G>2. Please double-check the data.
2) Please report the total dose received to the upper GI tract in patients who experienced G>2 GI acute/late toxicity
-Discussion: I think that the radiobiological evidence about the use of CIRT for pancreatic cancers as well as the management of organ motion/ anatomical variations intra and inter-fractions should be mentioned and improved
Author Response
Response to Reviewer 1 Comments
Thank you for your very constructive comments. We have revised the manuscript to respond your questions as follows.
-Patient eligibility:
1): please specify what is the minimum distance between the tumor and gastrointestinal tract
Response: The median distance was 3mm with range of 0-24mm. I added this information to table 1.
2) please specify if portal thrombosis is an exclusion criterion. If not, please detail what is the risk of liver failure due to portal thrombosis considering the recent data both in pancreatic cancers and HCC who underwent HCC. (doi: 10.1002/cncr.27723; Kasuya, doi: 10.18632/oncotarget.27028)
Response: We did not set portal vein thrombosis as an exclusion criterion. This is because the stenosis of the portal vein at the pancreatic part, due to factors such as thrombosis, is compensated by the development of collateral vessels, and the occurrence of liver failure due to this stenosis is extremely rare. The paper you provided (doi: 10.1002/cncr.27723) is a paper on preoperative CIRT for pancreatic cancer, and indeed there is grade 4 liver damage, but this is likely due to resection of the long segment of the superior mesenteric-splenic-portal vein confluence after CIRT. Previous publications on definitive CIRT for pancreatic cancer and our results in this study have not observed serious liver injury from causes other than biliary stricture, and we believe that the risk of liver failure from CIRT is low.
3) please specify why metal stent is an exclusion criterion
Response: Metal stents significantly affect the delivery distance of carbon ions, so cases with metal stents are not eligible. We added this information to the section of patient eligibility.
- Carbon ion radiotherapy
1) please detail the radiobiological model used
Response: We used the mixed beam model. We added the information.
2)do you use fixed beamlines or a gantry?
Response:: We used fixed beamlines. We added the information.
3) please specify the optimization of treatment that you used ( robust optimisation?
Response: In this study, we always perform a second respiratory-gated CT scan the day before the start of CIRT to confirm the dose distribution to the GTV and risk organs and, if necessary, to make a new treatment plan. Moreover, we have performed tumor location matching using in room CT, and if there was a large amount of displacement, a new treatment plan was created based on the new CT. We added the information.
4) How was the planned delivery gated?
Response: The treatment was delivered under respiratory gating. We added the information.
5) How did you check and also manage the organ motion and the inter/intra-fraction anatomical variation? (doi.org/10.1016/j.radonc.2022.09.005)
Response: We manage organ motion by adding a 2mm PRV margin and confirming the actual position of the organ by in-room CT. Moreover, we always perform a second respiratory-gated CT scan the day before the start of CIRT to confirm the dose distribution to the GTV and risk organs and, if necessary, to make a new treatment plan. In addition, we used in-room CT for position matching and if there was a large amount of displacement, a new treatment plan was created based on the new CT.
- Treatment planning:
1) please define the lymph node areas according to the localization of the tumour and the guidelines used
Response: As prophylactic lymph node areas, we included the celiac, superior mesenteric, peripancreatic, portal, and para-aortic regions cephalad to the transverse portion of the duodenum. We added this information.
2) please provide a table of the constraints used during the planning. In particular, considering the liver toxicity reported in CIRT literature I do think it is of utmost importance to specify the constraints used for portal vein, vessels, liver, and biliary tracts as well as for gastro-intestinal structures (colon, small bowel, duodenum, stomach…)
Response: The portal vein at the pancreatic portion is often obstructed by the tumor itself, but since it is compensated for by the development of collateral blood vessels, it is rarely a problem after CIRT, and we do not set dose constraints for portal vein. As for the bile ducts, since they are often compressed by the tumor in pancreatic head cancer and it is difficult to avoid irradiating them, thus, we do not set dose constraints for bile ducts. However, it is known that insertion of large diameter stents such as metal stents into the bile ducts after irradiation can cause severe adverse events such as bleeding and ulceration. Therefore, we strongly recommend avoiding the use of metallic stents after heavy particle irradiation, and because of this, no serious biliary tract disorders have appeared.
For the liver, the dose constraint is set so that the volume exceeding 20 Gy(RBE) is less than 35% of the total volume, and for the kidney, the volume exceeding 20 Gy(RBE) is less than 50% of the each kidney volume. We added this information into a new table 1.
-Chemotherapy
1) Please specify the criteria for which a patient received FORFIRINOX or GEM
Response: We set the criteria for chemotherapy administration were set as follows: white blood cells ≥ 2,000cells/mm3, neutrophils ≥ 1,000cells/mm3, platelets ≥ 75,000/mm3, blood bilirubin < 3.0mg/dL, serum aspartate aminotransferase and alanine aminotransferase < 200IU/L, and serum creatinine < 1.5mg/dL. We added them.
-Toxicities:
1) You reported “ acute grade 2 or higher ulcers were found in 4 patients”, but the table reported 4 G2 and no G>2. Please double-check the data.
Response: We have corrected as follows “Acute grade 2 ulcers in the upper gastrointestinal... found in 4 patients”
2) Please report the total dose received to the upper GI tract in patients who experienced G>2 GI acute/late toxicity
Response: We compared the V30 values in patients with and without GI ulcers and found no significant difference between the two groups as a result of the t-test. We have added this result to the manuscript.
-Discussion: I think that the radiobiological evidence about the use of CIRT for pancreatic cancers as well as the management of organ motion/ anatomical variations intra and inter-fractions should be mentioned and improved
Response: We have added as follows “However, in CIRT for the pancreas, it is necessary to manage the intra- and inter fraction movement of the GTV and risk organs. Recently, plan optimization with scheduled multiple CT imaging for re-evaluation has been reported to be useful in overcoming the intra- and inter fraction movement [doi.org/10.1016/j.radonc.2022.09.005]. We currently perform only a single re-evaluation CT the day before treatment begins, but are considering performing multiple CTs routinely as described in this report.”

Reviewer 2 Report
In this clinical study, the authors analyzed the clinical outcomes of Carbon Ion Radiotherapy (CIRT) for locally advanced pancreatic cancer in the systemic chemotherapy era. It is a retrospective and monocentric study carried out on 44 patients with locally advanced unresectable pancreatic cancer (URPC) whose were included between January 2016 and December 2020. All patients were treated with CIRT with 37 patients received neoadjuvant chemotherapy (NAC) The eligibility and exclusion criteria are well presented in the manuscript. The ethics regulations are respected: approbation by committee of human clinical research and all patients signed contentment. The main parameters statically evaluated are 2-year overall survival (OS), local control (LC), and progression- free survival (PFS) rates as well as adverse events that occurred after CIRT (acute and late toxicities).
The manuscript is very well written and clear. The results presented are interesting and innovative with a clear clinical relevance for the management of these patients. The authors show CIRT associated to systemic chemotherapy has survival benefits for URPC patients. This study has some limitations which are clearly exposed by the authors at the end of the manuscript (lines 305-311).
My main concern is the number of patients indicated in the figure 1b (44 patients) since only 37 patients were treated with NAC. Is it mistake? If so, it will have to be corrected in the figure and the text as well as the studied parameters and statistics before the article is published.
To support the treatment modalities by CIRT, a figure presenting the dosimetry maps and DVH analyses would be appreciable.
The overall manuscript is written in good quality English. Some errors present and the minor editing of English language is recommended.
Author Response
Response to Reviewer 2 Comments
Thank you for your very constructive comments. We have revised the manuscript to respond your questions as follows.
Comments:My main concern is the number of patients indicated in the figure 1b (44 patients) since only 37 patients were treated with NAC. Is it mistake? If so, it will have to be corrected in the figure and the text as well as the studied parameters and statistics before the article is published.
To support the treatment modalities by CIRT, a figure presenting the dosimetry maps and DVH analyses would be appreciable.
Response: Sorry for the missing words, the 7 cases who did not receive NAC were analyzed from the start date of treatment, i.e., the start date of CIRT. The number of patients is correct as 44, since this is the analysis of all patients. We have revised the manuscript and the description of Figure 1. In addition, we added a new figure of the dosimetry map.

Reviewer 3 Report
Authors report outcomes of C-ion RT for locally advanced PC based on a retrospective setup. While advancements in chemotherapy have prolonged the prognosis of these patients, outcomes are far from satisfactory. Overall, this manuscript is well written, and this study will provide the community with new insight regarding this matter.
I would like to point out a few minor things to maximize the impact of this report.
1. Title should be self-explanatory i.e., C-ion RT with concurrent chemotherapy for LA-PC: a retrospective case series analysis.
2. Throughout the manuscript, the use of “Gy (RBE)” has been depreciated in ICRU 93. Please comply.
3. In M&M eligibility: “1) pathologically proven locally advanced URPC” is virtually impossible. Do you mean unresectable pathologically proven PC? How was the respectability determined? Also, where there any cases other than ductal adenocarcinoma (ie acinar cell ca.)?
4. In M&M follow-up: every 6 months contradicts to what is written later. Was it really every 6 months? If so, the granularity of the data is too coarse for a patient dataset with MST less than 3 years let alone evaluation of acute A/E.
5. “S-1” is a registered trademark. While probably nullified due to lack of claim by the filer, please use Tegafur/Gimeracil/Oteracil
6. Please spell out “GnP” when using for the first time in the manuscript.
7. Various cut-off values are used in risk factor analysis. Where did these values come from? Please elaborate.
It appears so that the manuscript has been altered during or after the proofreading process leading to some inconsistency.
Author Response
Response to Reviewer 3 Comments
Thank you for your very constructive comments. We have revised the manuscript to respond your questions as follows.
Comment 1: Title should be self-explanatory i.e., C-ion RT with concurrent chemotherapy for LA-PC: a retrospective case series analysis.
Response: I corrected the title as “Carbon-ion Radiotherapy Combined with Concurrent Chemo-therapy for Locally Advanced Pancreatic Cancer: a retrospective case series analysis”
Comment 2: Throughout the manuscript, the use of “Gy (RBE)” has been depreciated in ICRU 93. Please comply.
Response: We have replaced “Gy(RBE)” with “Gy” and added that Gy for CIRT is expressed as RBE-weighted.
Comment 3: In M&M eligibility: “1) pathologically proven locally advanced URPC” is virtually impossible. Do you mean unresectable pathologically proven PC? How was the respectability determined? Also, where there any cases other than ductal adenocarcinoma (ie acinar cell ca.)?
Response: We intended to describe it as pathologically diagnosed pancreatic adenocarcinoma that was also diagnosed as unresectable by radiological imaging. Also, there was no case who had other than ductal adenocarcinoma. We have revised it.
Comment 4: In M&M follow-up: every 6 months contradicts to what is written later. Was it really every 6 months? If so, the granularity of the data is too coarse for a patient dataset with MST less than 3 years let alone evaluation of acute A/E.
Response: In fact, most cases were followed up every three months, but for those who lived far away and had difficulty coming to the hospital, visits were allowed every six months, so we used the description "follow-up at least every six months. We have changed the description to be more factual, stating that the patient is followed up every three months.
Comment 5: “S-1” is a registered trademark. While probably nullified due to lack of claim by the filer, please use Tegafur/Gimeracil/Oteracil
Response: We replaced “S-1” with Tegafur/Gimeracil/Oteracil.
Comment 6: Please spell out “GnP” when using for the first time in the manuscript.
Response: We revised it.
Coment 7: Various cut-off values are used in risk factor analysis. Where did these values come from? Please elaborate.
Response: To calculate the cutoff values for continuous variables, we basically use receiver operating characteristic (ROC) curve analysis. For example, in the ROC analysis for age, the youden index was maximum at age 69.5 years, so the cutoff value was set at 70 years. We have added how to set the cut-off value in the M&M section.

Round 2
Reviewer 1 Report
The authors significantly improved the manuscript. I suggest, however, to report in the text the answers (well explained in the author's response for the reviewers) about the portal vein and the absence of contraindications/constraints in the method and/or the discussion. It's an important point.
In particular, as reported by the Authors:
- We did not set portal vein thrombosis as an exclusion criterion. This is because the stenosis of the portal vein at the pancreatic part, due to factors such as thrombosis, is compensated by the development of collateral vessels, and the occurrence of liver failure due to this stenosis is extremely rare. The paper you provided (doi: 10.1002/cncr.27723) is a paper on preoperative CIRT for pancreatic cancer, and indeed there is grade 4 liver damage, but this is likely due to resection of the long segment of the superior mesenteric-splenic-portal vein confluence after CIRT. Previous publications on definitive CIRT for pancreatic cancer and our results in this study have not observed serious liver injury from causes other than biliary stricture, and we believe that the risk of liver failure from CIRT is low.
-The portal vein at the pancreatic portion is often obstructed by the tumor itself, but since it is compensated for by the development of collateral blood vessels, it is rarely a problem after CIRT, and we do not set dose constraints for portal vein
Author Response
Thank you for your very constructive comments. We have revised the manuscript to respond your questions as follows.
Comment:
The authors significantly improved the manuscript. I suggest, however, to report in the text the answers (well explained in the author's response for the reviewers) about the portal vein and the absence of contraindications/constraints in the method and/or the discussion. It's an important point.
Response: We have added the description in the method section as “The portal vein in the pancreatic portion is often obstructed by the tumor itself, but this is compensated by the development of collateral vessels. Thus, if stenosis occurs in this area after CIRT, it rarely causes serious liver damage. Therefore, we did not set dose constraints for the portal vein or use portal vein stenosis as an ineligibility criterion.”.
Moreover, the entire manuscript was proofread again in English by another expert. The English text should be more clear to understand.

Reviewer 2 Report
The corrections made by the authors have significantly improved the manuscript, and they have completely answered my questions raised in the previous revision. However, I noticed a word error in table 4: GTV is gross tumor volume and not “gloss”. The overall manuscript is written in good quality English. Some errors present and the minor editing of English language is recommended.
Some errors are present and the minor editing of English language is recommended.
Author Response
Thank you for your very constructive comments. We have revised the manuscript to respond your questions as follows.
Comments: The corrections made by the authors have significantly improved the manuscript, and they have completely answered my questions raised in the previous revision. However, I noticed a word error in table 4: GTV is gross tumor volume and not “gloss”. The overall manuscript is written in good quality English. Some errors present and the minor editing of English language is recommended.
Response: Thanks for pointing out the typo. It has been corrected. Moreover, the entire manuscript was proofread again in English by another expert. The English text should be more clear to understand.
